# *Momordica charantia* Extract Protects against Diabetes-Related Spermatogenic Dysfunction in Male Rats: Molecular and Biochemical Study

**DOI:** 10.3390/molecules25225255

**Published:** 2020-11-11

**Authors:** Gamal A. Soliman, Rehab F. Abdel-Rahman, Hanan A. Ogaly, Hassan N. Althurwi, Reham M. Abd-Elsalam, Faisal F. Albaqami, Maged S. Abdel-Kader

**Affiliations:** 1Department of Pharmacology, College of Pharmacy, Prince Sattam Bin Abdulaziz University, Al-Kharj 11942, Saudi Arabia; drgamal59@hotmail.com (G.A.S.); h.althurwi@psau.edu.sa (H.N.A.); f.albaqami@psau.edu.sa (F.F.A.); 2Department of Pharmacology, College of Veterinary Medicine, Cairo University, Giza 12211, Egypt; 3Department of Pharmacology, National Research Centre, Giza 12622, Egypt; Rehabs2001@yahoo.com; 4Department of Chemistry, College of Science, King Khalid University, Abha 61421, Saudi Arabia; ohanan@kku.edu.sa; 5Department of Biochemistry, College of Veterinary Medicine, Cairo University, Giza 12211, Egypt; 6Department of Pathology, College of Veterinary Medicine, Cairo University, Giza 12211, Egypt; rehammahmoudpathology@gmail.com; 7Department of Pharmacognosy, College of Pharmacy, Prince Sattam Bin Abdulaziz University, Al-Kharj 11942, Saudi Arabia; 8Department of Pharmacognosy, College of Pharmacy, Alexandria University, Alexandria 21215, Egypt

**Keywords:** *Momordica charantia*, diabetes, male fertility, Bax/Bcl-2, caspase-3

## Abstract

More than 90% of diabetic patients suffer from sexual dysfunction, including diminished sperm count, sperm motility, and sperm viability, and low testosterone levels. The effects of *Momordica charantia* (MC) were studied by estimating the blood levels of insulin, glucose, glycosylated hemoglobin (HbA1c), testosterone (TST), follicle-stimulating hormone (FSH), and luteinizing hormone (LH) in diabetic rats treated with 250 and 500 mg/kg b.w. of the total extract. Testicular antioxidants, epididymal sperm characteristics, testicular histopathology, and lesion scoring were also investigated. Testicular mRNA expression of apoptosis-related markers such as antiapoptotic B-cell lymphoma-2 (Bcl-2) and proapoptotic Bcl-2-associated X protein (Bax) were evaluated by real-time PCR. Furthermore, caspase-3 protein expression was evaluated by immunohistochemistry. MC administration resulted in a significant reduction in blood glucose and HbA1c and marked elevation of serum levels of insulin, TST, and gonadotropins in diabetic rats. It induced a significant recovery of testicular antioxidant enzymes, improved histopathological changes of the testes, and decreased spermatogenic and Sertoli cell apoptosis. MC effectively inhibited testicular apoptosis, as evidenced by upregulation of Bcl-2 and downregulation of Bax and caspase-3. Moreover, reduction in apoptotic potential in MC-treated groups was confirmed by reduction in the Bax/Bcl-2 mRNA expression ratio.

## 1. Introduction

Diabetes is a long-term metabolic disorder that represents a great public health issue as a result of present lifestyle and dietary practices. The prevalence of diabetes has accelerated quickly. World Health Organization (WHO) statistics expect that the number of people with diabetes will reach 366 million by 2030 [1].

Several reports from studies on experimental animals and men with diabetes showed that diabetes results in reproductive complications, as high glucose may lead to oxidative stress and cell apoptosis [2]. Altered sperm morphology and reduced sperm motility and counts are the main factors implicated in the decreased fertility of men with diabetes [3] and animal models [4]. Additionally, spermatogenic failure is one of the common adverse effects of diabetes [5]. It is unclear whether the injury is due to local effects of increased blood glucose levels or by changes in hormone levels that damage the hypothalamic–pituitary gonadal axis [6].

Current therapeutic measures to treat diabetes include the use of insulin and other agents, such as amylin analogues, alpha glycosidase inhibitors, sulfonylureas, and biguanides. These drugs have certain adverse effects, such as hypoglycemia at higher doses, liver problems, lactic acidosis, and diarrhea. In addition to these available therapeutic options, many herbal medicines have been recommended for the management of diabetes. Ethnobotanical information suggests that approximately 800 plants may have significant antidiabetic effects [7].

For the present study, *Momordica charantia* (Cucurbitaceae) (MC) was chosen since it is the most greatly inspected and most widely acclaimed remedy for the management of diabetes since ancient times [8]. *M. charantia*, also referred to as bitter melon, bitter gourd, or karela, is a member of the Cucurbitaceae family and is commonly used as a traditional remedy for diabetes in India, Asia, Africa, and South America [9]. *M. charantia* is a widely used and extensively studied herb for its glycemic control effect in diabetes [10]. Many clinical trials have also confirmed its hypoglycemic action [11].

The fruit of *M. charantia* has been used as a vegetable in India. Other pharmacological properties of MC, such as antioxidant, anti-inflammatory, hepatoprotective, antibacterial, anthelmintic, antiviral, antitumor, and antiulcer activities, have been reported [12]. In Turkish folk medicine, mature fruits of the plant are used topically for the treatment of wounds and internally for intestinal ulcers. In Indian folk medicine, different preparations of the plant are used for eczema, leprosy, dysmenorrhea, pneumonia, gout, psoriasis, jaundice, kidney stones, rheumatism, and scabies. Several phytochemicals of MC fruits have been reported in the literature. These phytochemicals include flavonoids [13], polysaccharides, saponins and phenolic compounds [14], triterpenoids [15], alkaloids [16], and sterols [17].

The development of diabetic complications is strongly related to chronic sustained hyperglycemia. Further, oxidative stress has been reported as a major pathway in the pathogenesis of diabetic complications [18]. Consequently, the antidiabetic and antioxidant activities of MC may successfully stop the development of diabetic complications. Hence, this study was aimed at assessing the potential effect of *M. charantia* extract on protection against diabetes-related sexual complications in male rats.

## 2. Results

### 2.1. LC-MS Analysis

LC-MS analysis enables semiquantitative identification and estimation of major phytochemicals such as momordicin **I** and **II** (Figure 1) based on the peak areas of each compound [19].

### 2.2. Effects on Blood Levels of Insulin, FBG, and HbA1c

On the day of diabetes induction, no marked differences were observed in serum insulin and glucose levels among all groups of rats. Three days after STZ injection, blood of all diabetic groups presented a marked reduction in insulin and remarkable elevation in glucose in comparison to the NC group, although there were no differences among them. At the end of the experiment, MC-250-treated groups presented a remarkable reduction (*p* ≤ 0.05) in FBG and increase (*p* ≤ 0.05) in insulin levels compared to the DC rats (Table 1). The high dose of MC was more efficacious and was comparable to the NC group in controlling blood levels of glucose and insulin in diabetic rats after 12 weeks of treatment.

DC rats displayed a significant increase in HbA1c % in their blood compared to the NC group (*p* ≤ 0.05). After medication with MC-250 or MC-500, the percentage of HbA1c resettled toward the normal control value. In parallel with the improvement of blood glucose concentration, the significant reduction in blood HbA1c in animals treated with MC-500 was comparable to that observed in the NC group.

### 2.3. Effect on Antioxidant Stress Markers in Testicular Tissue

The oxidative impact of diabetes on the testicular tissue was evaluated by determining the activity of enzymes such as SOD, GPx, and CAT (Table 2). The results showed that STZ-induced diabetes reduced the antioxidant enzyme activity in the testicular tissue compared with the NC rats (*p* ≤ 0.05). In addition, the GSH level in the testicular homogenate of DC rats was significantly reduced compared to the NC rats. It was observed that STZ-induced diabetes increased the level of MDA, the major end product of lipid peroxidation (LPO), in the testicular tissues compared with the NC group. This investigation showed a remarkable elevation in SOD, GPx, and CAT activity and GSH content in the testicular tissues of D+MC-250- and D+MC-500-treated rats compared to the DC group (*p* ≤ 0.05).

When the diabetic rats were given MC-250 and MC-500 for 12 weeks, reductions in MDA levels were observed in their testicular tissues compared with the DC group. Additionally, both doses of MC significantly (*p* ≤ 0.05) improved the testicular activity of SOD, GPx, and CAT enzymes. Interestingly, D+MC-500 was able to maintain the levels of oxidative enzymes and LPO in testicular tissue similar to those of the NC group.

### 2.4. Effect on Serum Levels of TST and Gonadotropins

As shown in Table 3, STZ-induced diabetes produced a significant decrease in serum TST, FSH, and LH in the DC rats compared with the NC group. Administration of MC-250 and MC-500 to diabetic rats produced significant increases in the blood levels of TST, FSH, and LH compared to the DC group (*p* ≤ 0.05), however, the hormonal concentration did not restore to normalized control values.

### 2.5. Effect on Sperm Characteristics

Effects of MC-250 and MC-500 treatment on sperm characteristics of male rats are illustrated in Table 4. Sperm count was less in DC rats than in NC rats. Administration of MC-250 and MC-500 for 12 weeks caused a significantly higher count compared to DC rats. Meanwhile, sperm motility in the DC group was lower than that in NC rats. Medication with MC-250 and MC-500 for 12 weeks resulted in markedly improved sperm motility compared to DC rats. Sperm viability was lower in DC rats than NC rats. Administration of MC-250 for 12 weeks resulted in a significantly improved viability percentage compared to DC rats, whereas treatment with MC-500 resulted in a higher viability percentage.

Semen analysis for morphological structure of sperm (Table 4) revealed a significantly higher percentage of morphologically abnormal spermatozoa in the DC group than the NC group. The predominant types of abnormalities were alterations of the tail, such as coiled, bent, and broken tails, and middle piece, such as bent middle piece (Figure 2). Administration of MC at 250 and 500 mg/kg b.w. for 12 weeks significantly reduced the percentage of abnormal-shaped sperm cells compared to the DC group.

### 2.6. Effect on Body Weight and Relative Weight of Sexual Organs

As shown in Table 5, body weight at the beginning of the experiment did not differ between the nondiabetic and diabetic groups of animals. By the end of the 12 weeks, there was a significant decrease in the weight of DC rats compared to the NC group. Treatment with MC-250 and MC-500 significantly increased the final body weight in diabetic animals compared with the DC rats. Interestingly, the increment in body weight of D+MC-500 rats was found to be nonsignificant when compared with the normal control group.

The relative weights of the reproductive organs were markedly lower in the DC group compared with the normal control group (Table 6). The relative weights of the reproductive organs were remarkably increased in diabetic animals treated with MC-250 and MC-500 compared with the DC group. Interestingly, the animals of the D+MC-500 group had higher relative weight of the reproductive organs than all other groups, which did not significantly differ from that of the nondiabetic group.

### 2.7. Histopathological Examination of Testis

The normal untreated control group showed unbroken seminiferous tubules, with intact spermatogenic proliferation of cells and sperms (Figure 3A). Diabetic control rats showed alterations in the normal histological architecture of the testes because of a high level of blood glucose. They showed severe testicular damage with significant reduction in the number of spermatogonial cells and primary and secondary spermatocytes and massive numbers of spermatid giant cells. In addition, large number of seminiferous tubules contained only one or two layers of vacuolated Sertoli cells with complete disappearance of spermatogenic series (Figure 3B,C). Moreover, intense intertubular edema was also detected.

However, D+MC-250 and D+MC-500 showed improved histology of the testis, with many intact tubules and spermatozoa-filled lumen (Figure 3C–F). The seminiferous tubules were active with normal main cells. Figure 4A shows the testicular lesion scoring of the different groups.

### 2.8. Immunohistochemistry of Caspase-3

Figure 4B shows the immunohistochemical evaluation of anti-caspase-3 protein expression expressed as area percentage. Anti-caspase-3 protein expression was localized in cytoplasm and nucleus of spermatogonial cells, primary spermatocytes, secondary spermatocytes, spermatid, and Sertoli cells. The normal control group showed very weak anti-caspase-3 protein expression in all seminiferous tubules (Figure 5A). The DC group showed a significant elevation of anti-caspase-3 protein expression compared to the NC group (Figure 5B). However, diabetic rats exposed to MC 250 and 500 mg/kg b.w. showed a significant reduction in anti-caspase-3 protein expression (Figure 5C,D) compared with the DC group.

### 2.9. Relative mRNA Expression Levels of Apoptosis-Related Genes

RT-PCR was performed to investigate the impact of MC on the mRNA expression of Bax and Bcl-2 genes in testicular tissue of diabetic rats. Moreover, the apoptotic tendency of testicular tissues was evaluated via the Bax mRNA to Bcl-2 mRNA ratio. The relative Bax mRNA expression in the testes of DC rats was significantly higher (Figure 6A), while the relative Bcl-2 mRNA expression was remarkably lower in comparison with NC rats (Figure 6B). Thus, STZ-induced diabetes caused a significant elevation in the Bax/Bcl-2 mRNA ratio in the testis compared to NC (Figure 6C). Administration of MC-250 and MC-500 significantly decreased the transcriptional expression of Bax and upregulated Bcl-2 expression (*p* ≤ 0.05) in the testicular tissues of diabetic rats compared with the DC group. Furthermore, the Bax/Bcl-2 ratio was significantly decreased in response to MC treatment.

### 2.10. Male Fertility Test

The reproductive ability of male rats with diabetes was reduced, as indicated by the decreased pregnancy rate (33.3%), number of viable fetuses per rat (5.7 ± 0.27), and decreased male fertility percentage (64.77%) compared with NC rats (91.6%, 12.2 ± 0.56, and 92.42%, respectively). The results presented in Table 7 show that oral administration of MC-250 and MC-500 for 12 weeks caused a significant (*p* ≤ 0.05) increase in pregnancy rates, number of viable fetuses per rat, number of corpora lutea (CL), and percentage male fertility.

## 3. Discussion

A semiquantitative LC-MS study of MC extract resulted in the identification of stearic acid (8.34%) and oleic acid (4.28%) as the most abundant components. However, the most important components relative to the antidiabetic effect were the steroidal aglycone momordicin **I** (1.73%) and its glucopyranosyl derivative momordicin **II** (1.37%) [12,19].

Many reports indicate that MC extract can be used as therapy for the management of diabetes [20]. In the present investigation, DC rats showed decreased insulin levels and elevated FBG and HbA1c levels, which are characteristics of hyperglycemia. Further, administration of MC ameliorated the altered levels of insulin, FBG, and HbA1c in blood. This suggests that MC is able to improve the ability of insulin to reduce blood glucose, confirming its antidiabetic activity. In a previous study, aqueous extract of MC fruits at a dose of 20 mg/kg b.w. for 4 weeks was found to reduce FBG of diabetic rats by 48%, an effect comparable to that of glibenclamide at 0.1 mg/kg b.w. [9]. Other studies also reported that the antidiabetic effect of MC was comparable with tolbutamide [21,22] and chlorpropamide [23]. The elevated insulin level following MC medication suggests an insulin secretagogue mechanism of action. Further, the results of some in vitro studies suggested that MC stimulates insulin secretion from isolated pancreatic islet cells [24,25]. Earlier studies by Mohammady et al. [26] indicated that MC reduced intestinal glucose absorption in diabetic rats.

Diabetes-induced hyperglycemia is known to induce the generation of reactive oxygen species (ROS), which are involved in testicular oxidative stress [27]. Increased levels of ROS induced apoptosis of the testicular germ cells of diabetic rats, leading to degeneration of the testes [28]. In this investigation, MDA content increased and SOD, GPx, and CAT activity decreased in the testicular tissue of DC rats, indicating that the tissue was under oxidative stresses. MC significantly decreased LPO in diabetic rats, as evidenced by the reduced testicular MDA content. Meanwhile, MC treatment also increased SOD, GPx, and CAT activity in the testes of diabetic rats. These results therefore show the possible efficacy of MC in improving the antioxidant defense system in the testicular tissue of diabetic rats. MC is thought to protect against many disorders due to the presence of phytochemicals with antioxidant potential [29]. The potential mechanism of MC as an antioxidant might be due to its flavonoid content. Hanasaki et al. [30] found that flavonoids are potent radical scavengers, as they stabilize the ROS by reacting with the reactive compound of the radical. Further, the high levels of polysaccharides, saponins, and phenolic compounds in MC indicate its protective effect against oxidative stress [14].

We also observed that LPO was remarkably accumulated in the testes of DC rats. A reduced level of insulin stimulates the activity of fatty acyl coenzyme A oxidase, which results in oxidation of fatty acids and lipid peroxidation [31]. LPO reduces the fluidity of cellular membranes and alters the interaction between membrane-bound enzymes and receptors, resulting in cellular injury. MC medication significantly reduced the level of LPO in the testes of diabetic rats.

Reproductive hormones such as TST and gonadotropins (FSH and LH) are believed to be major biomarkers to evaluate male reproductive function. It was reported that the release of TST by Leydig cells is enhanced by LH. Further, LH stimulates FSH to bind with Sertoli cells to stimulate the process of spermatogenesis [32]. In the current study, serum levels of TST and gonadotropins were markedly decreased in DC rats. Low TST levels and testicular dysfunction have also been documented in male animals and men with diabetes [33]. The reduced serum level of TST in DC rats might be an effect of glycemia on the function of Leydig cells and/or due to oxidative stress induced by diabetes [15].

In fact, an association between insulin, gonadotropins, and TST has been described. A low level of blood insulin blocks secretion of gonadotropins, thus reduces the release of TST in the testicular tissue [34]. Fortunately, administration of either dose of MC increased the levels of TST, FSH, and LH in comparison with DC values. Hyperglycemia might influence serum testosterone levels in men with poorly controlled type 2 diabetes [35]. In a study of such men, their testosterone levels were found to be negatively correlated with FBG and HbA1c values. Therefore, the increase in serum levels of TST, FSH, and LH in MC-treated rats could be attributed to the antioxidant and antidiabetic properties of the phytochemical contents of MC, which can counteract free radicals.

Altered sperm characteristics were widely observed in diabetic male animals [36]. Sperm count, motility, and morphology are considered as important markers to assess the effects of chemicals and extracts on sperm characteristics. Spermatogenesis and TST levels are lower in humans with diabetes compared to healthy individuals [36]. Diabetes disrupts sperm characteristics by mechanisms that involved ROS production and LPO generation. Excessive production of ROS interferes with the mitochondrial membrane potential and reduces energy availability, which may block sperm motility [37].

In this investigation, administration of MC resulted in higher sperm count and motility. Increased blood glucose levels have been correlated with altered sperm quality [38]. In addition, reduced sperm quality is correlated with decreased TST levels [39]. Further, increased LPO has been correlated with disturbed sperm membrane and reduced sperm motility and fertilization potential [40]. Therefore, the improvement of sperm characteristics in diabetic rats exposed to MC can be attributed to the increased serum level of TST and the ability of the extract to protect against hyperglycemia and increased LPO. The percentage of viable sperm was also reduced in the DC rats. This observation is consistent with previous reports on rodents [41] and humans [42]. In diabetes, injury of the seminiferous tubules and increasing sloughing of immature germ cells result in abnormal sperm development [43]. Meanwhile, the ability of MC to ameliorate blood glucose and HbA1c levels of diabetic rats could also contribute to reducing the chance of acquiring abnormal spermatozoa and sperm oxidative stress [44].

The findings of sperm characteristics were confirmed by the histopathological and immunohistochemical results. In the present study, induction of diabetes in rats resulted in severe testicular degeneration with marked reduction in the numbers of spermatogenic and Leydig cells with vacuolation of Sertoli cells and intertubular edema. Diabetes has been reported to stimulate cell apoptosis and seminiferous tubule degeneration, which are markers for the impairment of spermatogenesis [2]. Additionally, Kim et al. [35] reported that suppression of TST production by Leydig cells resulted in germ cell apoptosis through DNA fragmentation, and subsequent marked reduction in number of spermatocytes and spermatids [35]. Treatment of diabetic rats with either dose of MC improved STZ-induced diabetic testicular histological alterations, with restoration of normal architecture of seminiferous tubules and slight reduction in spermatogenic cells. The testicular protective effect of MC may reflect its antioxidant nature as well as its basic role in sperm maturation [45].

Diabetes-induced weight loss and atrophy of male reproductive organs have been reported to be induced by oxidative stress and apoptosis [46]. The current investigation shows a significant reduction in the weight of male reproductive organs of the DC group in comparison with the NC group, indicating that weight loss of reproductive organs may be one of the reasons for their dysfunction. Administration of MC to diabetic rats recovered the weight of their reproductive organs toward the NC level, which could be explained by its antidiabetic and antioxidant properties. Further, physiological growth and function of male reproductive organs depend on hormones such as TST [47]. Therefore, the improvement in the weight of reproductive organs in D+MC-medicated groups could be related to the increased level of TST in their blood.

At the molecular level, testicular failure in the diabetic condition was confirmed by a genomic study of the testicular tissue. Previous reports have proved that apoptosis of the testicular cells plays a critical role in testis dysfunction in diabetic rats [48]. In the current study, two valuable biomarkers, Bax, a proapoptotic marker, and Bcl-2, an antiapoptotic marker, were evaluated. The antiapoptotic Bcl-2 is a negative regulator of cell death, inhibiting cells from undergoing apoptosis induced by different stimuli, whereas the proapoptotic Bcl2-associated X protein (Bax) blocks the antiapoptotic proteins, thus enhancing apoptosis [49]. The Bax/Bcl-2 ratio is considered as an index for cell susceptibility to apoptosis [50]. In diabetic rats, Bax gene expression was upregulated and Bcl-2 gene expression was downregulated due to the excessive production of ROS, which is responsible for germ cell apoptosis [51].

In the current study, elevated Bax and decreased Bcl-2 expression in testicular cells of DC rats indicated increased apoptotic processes. These results were compatible with previous studies [52]. Medication of diabetic rats with MC significantly improved the equilibrium of the proapoptotic (Bax) and antiapoptotic (Bcl-2) molecules. MC downregulated the expression of Bax and upregulated the expression of Bcl-2, thus effectively protecting against apoptosis of the testicular cells. Further, the ratio of Bax/Bcl-2 mRNA expression was increased in the testicular tissues of DC rats, indicating hyperglycemia-induced testicular apoptosis. The current results demonstrate that MC treatment suppressed apoptotic potential, as evidenced by its ability to decrease the Bax/Bcl-2 expression ratio. This, in turn, provides evidence for the antiapoptotic properties of MC.

Moreover, a high level of ROS within mitochondria signals the release of cytochrome c, which leads to caspase activation and apoptosis [53]. Caspase-3, a sort of cysteine protease, is an important mediator of apoptosis. Many reports have indicated that caspase-3 is upregulated in diabetic conditions [54]. In this study, we show that there was an association between diabetic sexual dysfunction and caspase-3 expression, as evidenced by apparent downregulation of caspase-3 content in the germ cells of D+MC-treated rats, suggesting that inhibition of caspase-3 activity may be one of the mechanisms by which MC protects against diabetic testicular injury.

In this investigation, the remarkable reduction in the percentage of mating success in DC rats indicates that their sexual desire was reduced by diabetes. Previous experimental studies reported that diabetes decreased male fertility through increased ROS production [55] and/or reduced number and motility of spermatozoa [56]. Further, the male fertility test revealed that MC-250 increased the mating success of diabetic males compared with DC rats, though the effect was less than that induced by MC-500. The increased mating success observed in the D+MC-500-treated group is a reflection of the recovery of erectile efficiency and the capability to achieve effective copulation by increasing serum levels of TST [57]. In male fertility tests, usually normal female rats are mated with treated males. The present results also show that the administration of MC to male rats increased their ability to impregnate females. A high number of fetuses carried by females impregnated by MC-treated male rats can be considered as evidence of successful fertilization. This effect can be attributed to increased sperm count and sperm motility; Donnelly et al. [58] reported that number and motility of sperm are directly correlated with effective fertilization and successful pregnancy [58].

The experimental doses used in our study (250 and 500 mg/kg b.w.) can be applied for human treatment. To convert the dose used in a rat to a dose based on surface area for humans, the rat dose was multiplied by the Km factor (6) for a rat and then divided by the Km factor (37) for a human [59]. This calculation results in human equivalent doses for MC of 40.54 and 81.08 mg/kg b.w., respectively, which equates to 2.43 and 4.86 g doses of MC for a 60 kg person.

One of the limitations of the current study is that we did not use a nondiabetic control group treated with the extract, as we were satisfied with comparing the results of the study with the normal control and diabetic control groups. The other limitation of the study includes the use of only two doses of MC (250 and 500 mg/kg b.w.), thus lacking the minimum dose useful for the clinical purpose. However, the strength of the current study was achieved by treating diabetic rats with MC for 12 weeks, which is an appropriate and well-used model for preclinical studies. Moreover, we studied various aspects of the diabetes-related male sexual complications by assessing the testicular antioxidants, estimating the level of sex hormones, examining the epididymal sperm characteristics, in addition to the histopathology and lesion scoring of the testis. In addition, testicular mRNA expression of apoptosis-related markers such as Bcl-2 and Bax was evaluated.

## 4. Materials and Methods

### 4.1. Plant Material and Extraction

Fresh fruits of *Momordica charantia* L. were purchased from the local market in Al-Kharj city, Saudi Arabia. The identity of the fruits was confirmed by Dr. Mohammad Atiqur Rahman, a taxonomist at the Medicinal, Aromatic and Poisonous Plants Research Center (MAPPRC), College of Pharmacy, King Saud University, Riyadh, Saudi Arabia. A voucher specimen (#16294) was deposited in the center. Five kilograms of fresh fruits were extracted by 90% ethanol at room temperature to yield 163.10 g of total extract [19]; conditions used for LC-MS analysis in positive and negative ion modes were described earlier [19].

### 4.2. LC-MS Study of the Extract

ESI-MS in positive and negative modes was carried out using an Xevo TQD Triple Quadrupole Mass Spectrometer (Waters Corporation, Milford, MA, USA). LC was performed on an ACQUITY UPLC-BEH using C18 column and water/methanol gradient containing 1% formic acid. More details were described earlier [19].

### 4.3. Experimental Animals

The Animal House Colony at the National Research Centre (NRC), Egypt, was the source of male Wistar rats (190 ± 20 g, 8 weeks of age). Animals were adapted to the standard laboratory environment for 1 week before the start of the study at 26 ± 2 °C, 12 h day/night cycle. Animals were given tap water ad libitum and a standard diet. The study protocol, including animal handling, was developed according to the protocol approved by the Institutional Animal Care and Use Committee, Cairo University (approval number: CU-II-F-14-18), and following the recommendations of the National Institutes of Health Guide for Care and Use of Laboratory Animals (Publication No. 85–23, revised 1985).

### 4.4. Induction of Experimental Diabetes

For diabetes induction, streptozotocin (STZ) (Sigma-Aldrich Corp., St. Louis, MO, USA) was injected intraperitoneally at 60 mg/kg b.w. [60] as solution in 0.01 M citrate buffer (pH 4.5). For the normal control group, only citrate buffer was injected. Fasting blood glucose (FBG) levels over 250 mg/dL in blood samples obtained from the tail veins of rats collected after 72 h as measured by a blood glucose meter (Accu-Chek Performa, Roche Diagnostics, Mannheim, Germany) were considered diabetic and utilized in our study.

### 4.5. Experimental Design

After being acclimatized for 1 week, the rats were randomly divided into 4 groups (n = 6). Groups 1 and 2, normal control (NC) and diabetic control (DC) groups, respectively, were treated with vehicle (3% Tween 80, 5 mL/kg). Groups 3 (D+MC-250) and 4 (D+MC-500), diabetic rats, were treated with MC at 250 and 500 mg/kg b.w., respectively. The selected doses of MC were based on our previous study [19]. MC extract was administered orally via oral tube for 12 weeks [61]. Initial body weight and blood levels of glucose, insulin, glycosylated hemoglobin (HbA1c), TST, and gonadotropins were quantified.

At the end of the experimental period, animals were fasted overnight, weighed, and euthanized by cervical dislocation, and 2 blood samples were collected from each rat through cardiac puncture. The first sample was collected into a tube containing EDTA for HbA1c estimation. The second sample was allowed to coagulate and then centrifuged at 3000 rpm for 15 min. Clear serum was collected and subjected to glucose, insulin, and hormonal assay.

### 4.6. Biochemical Estimation

Serum levels of glucose were estimated using Spinreact kits (Girona, Spain), and insulin levels were determined by enzyme-linked immunosorbent assay (ELISA) using a commercial kit (Cobas ELISA kits, Machelen, Belgium). Additionally, levels of TST and gonadotropins (FSH and LH) were measured using rat ELISA assay (Cobas ELISA kits, Machelen, Belgium). Blood levels of HbA1c were evaluated using QCA kits (Spain). All measurements were done following the manufacturer’s instructions.

### 4.7. Assessment of Oxidative Stress Markers in Testicular Tissues

Part of the left testis of each rat was homogenized (Heidolph Diax 900 homogenizer, Schwabach, Germany) in cold phosphate buffer saline followed by centrifugation at 3000 rpm for 10 min. The supernatants were collected and stored at −80 °C. The antioxidant enzyme activity, including superoxide dismutase (SOD), glutathione peroxide (GPx), and catalase (CAT), and the levels of reduced glutathione (GSH) and malondialdehyde (MDA) were estimated in the testicular homogenates with corresponding assay kits obtained from Cayman Chemical Company (Ann Arbor, MI, USA), following the procedures stated in the manufacturer’s instructions.

### 4.8. Assessment of Sperm Characteristics

#### 4.8.1. Sperm Motility

Cauda epididymis was cut into small pieces and incised to release epididymal fluid on a clean glass slide. Exactly, 2 μL of the epididymal fluid was mixed with 20 μL of 2.9% sodium citrate and cover slipped. The percentage of motile spermatozoa was evaluated under a light microscope (Hund Wetzlar H600/12, Wetzlar, Germany) at 400× magnification. Two samples from each rat were examined for sperm motility by two independent observers. At least 200 spermatozoa for each sample in at least two microscopic fields were estimated and categorized as motile or nonmotile.

#### 4.8.2. Sperm Counts

Counts of epididymal spermatozoa were performed according to the reported method [62]. Briefly, 200 μL of the epididymal fluid was added to a microtube containing 800 μL of normal saline. The tube contents were mixed well and transferred to an improved Neubauer counting chamber. The chamber was observed microscopically (Kruss MBL2000, A. Kruss Optronic, Hamburg, Germany) under magnification of 40×.

#### 4.8.3. Sperm Viability

Viability of spermatozoa was evaluated by adding one drop of eosin–nigrosin stain to a drop of epididymal fluid in an Eppendorf tube. After mixing, a drop of the stained epididymal fluid was placed on a glass slide and covered with coverslip. At least 200 sperm were observed under light microscope (Hund Wetzlar H600/12, Wetzlar, Germany) at 400× magnification. Dead sperm were stained red, while live ones were not stained. The percentage of sperm viability was calculated as (number of live spermatozoa/total spermatozoa counted) × 100.

#### 4.8.4. Sperm Morphology

One drop of epididymal fluid was mixed with 2 drops of eosin–nigrosin stain at one end of a clean warm microscopic glass slide. The mixed drops were drawn out with the edge of another slide that served as a spreader so that a thin film was formed and air-dried. The smears were examined under light microscope (Hund Wetzlar H600/12, Wetzlar, Germany) at 40× magnification, equipped with a digital camera (Canon EOS 550D). A minimum of 200 sperm were observed for morphological alterations. Sperm were classified into normal and abnormal. Sperm abnormalities were presented as percentage incidence.

### 4.9. Weights of Reproductive Organs

Immediately after rats were euthanized, testicles, epididymis, seminal vesicle, and prostate were dissected and individually weighed (Analytical Balance; Shimadzu AUW220D; Kyoto, Japan; range: 0.1 mg to 220 g). The relative weights of these organs were calculated as (organ weight/body weight) × 100.

### 4.10. Histopathological Examination

The testicles were obtained from the animals of the different groups and fixed in Bouin’s solution for 24 h. The tissue specimens were routinely processed to obtain 3–4 µm thick paraffin-embedded tissue sections. Tissue sections were stained with hematoxylin and eosin (H&E) stain following the method of Bancroft and Gamble (2008) [63]. A testicular scoring system was used according to the procedure reported earlier [64]. Thirty randomly selected seminiferous tubules were examined and evaluated in each experimental group. The score ranged from 10 to 1 according to presence of the main spermatogenic cells (spermatogonial cells, primary spermatocytes, secondary spermatocytes, and spermatid cells) and Sertoli cells.

### 4.11. Immunohistochemistry of Caspase-3

The immunohistochemistry of caspase-3 was evaluated according to a method described elsewhere [65]. Tissue samples were deparaffinized and rehydrated; then, an antigenic retrieval procedure was performed by treating the tissue sections with citrate buffer (pH 6) for 20 min. The sections were blocked by blocking solution to decrease the nonspecific background. The sections were incubated with rabbit anti-caspase-3 polyclonal antibody (ab13847; Abcam, Cambridge, UK) at 1:50 dilution overnight in a humidified chamber; then, the sections were washed 3 times with Tris-buffered saline. Goat Anti-Rabbit IgG H&L (HRP) (ab205718; Abcam, Cambridge, UK) was incubated with the specimens for 1 h, followed by incubation with DAB substrate (Sigma) to visualize the reaction. Tissue sections were counterstained with Mayer’s hematoxylin, then mounted, and the obtained images were analyzed by ImageJ Analyzer. Five sections from each group were examined and the area percentage of dark brown color was analyzed in 10 microscopic fields/slide.

### 4.12. Gene Expression Analysis

#### 4.12.1. Total RNA Extraction and cDNA Synthesis

Total RNA was extracted from the deep-frozen testis samples (~100 mg) using the phenol-based method (TRIzol^®^ Reagent, Qiagen, Hilden, Germany) according to the instructions provided. RNA quantity and purity were determined by NanoDrop spectrophotometer (Thermo Fisher Scientific, Waltham, Massachusetts, USA). Extracted RNA samples were subjected to reverse transcription using SuperScript III reverse transcriptase kit in the presence of oligo-dT primer [64].

#### 4.12.2. Quantitative Real-Time PCR

Testicular Bax and Bcl-2 mRNA expression levels were detected by quantitative RT-PCR using SYBR Green Master Mix (Thermo Fisher Scientific). PCR amplification reaction (20 µL) included 1 µL cDNA (25 ng), 10 µL SYBR Green, and 1 µL (5 pmol) of each primer for Bax (forward: 5′-ACCAAGAAGCTGAGCGAGTG-3′, reverse: 5′-CCAGTTGAAGTTGCCGTCTG-3); Bcl-2 (forward: 5′-GAGGATTGTGGCCTTCTTTG-3′, reverse: 5′-CGTTATCCTGGATCCAGGTG-3′); and β-actin (forward: ATGGTGGGTATGGGTCAG, reverse: CAATGCCGTGTTCAATGG) [66]. The PCR thermal program comprised predenaturation at 95 °C for 5 min and 40 cycles of amplification (denaturation at 95 °C for 10 s, annealing at 56 °C for 15 s, and extension at 72 °C for 20 s). Each assay was done in triplicate for each cDNA with a minus-template negative control. β-actin gene was amplified during the same reactions to serve as an endogenous control to normalize the Ct values of the target genes. The 2−∆∆Ct method was used to calculate the relative quantitative gene expression of Bax and Bcl-2 genes. Results are expressed as fold change compared with the NC group.

### 4.13. Male Fertility Test

Separately, 4 groups of male rats (n = 6) were used to assess their capability to fertilize females. The grouping of animals and dosing system were similar to those stated, with a protective effect against diabetes-related male sexual complications. They were orally treated with MC for 12 weeks. On the last day of the experimental period, each male rat was paired with 2 proestrus females (at 4:30–5:00 p.m.) overnight. The next morning (8:00–8:30 a.m.), effective mating was confirmed by observing sperm in the vaginal smears of female rats under a microscope. Mating success was calculated as (number of sperm-positive females/number of females paired) × 100. Sperm-positive females were isolated for 14 days, then euthanized with a high intraperitoneal dose of pentobarbital sodium and examined for confirmation of pregnancy. The numbers of corpora lutea and intrauterine fetuses were counted. The percentage of male fertility was calculated as (umber of fetuses/number of corpora lutea) × 100 [67].

### 4.14. Statistical Analysis

All obtained data were presented as mean ± SEM. Statistical analysis was done using one-way analysis of variance (ANOVA) followed by Tukey’s test to determine intergroup variability using Graph Prism^®^. A probability level of less than 0.05 was accepted as statistically significant.

## 5. Conclusions

In conclusion, the findings of the current investigation provide evidence that MC attenuates diabetes-associated sexual complications in male rats. The data also suggest that MC protects against male sexual complications through various mechanisms including improvement of hyperglycemia, activation of the antioxidant system, inhibition of LPO, reduction in Bax and increment of Bcl-2 at the gene and protein expression level, and maintenance of sperm quality. The study indicates that MC is a possible new protective agent for diabetes-related male sexual complications.

## Figures and Tables

**Figure 1 molecules-25-05255-f001:**
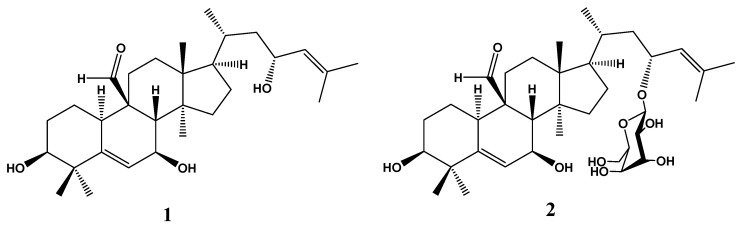
Chemical structure of momordicin I (**1**) and momordicin II (**2**).

**Figure 2 molecules-25-05255-f002:**
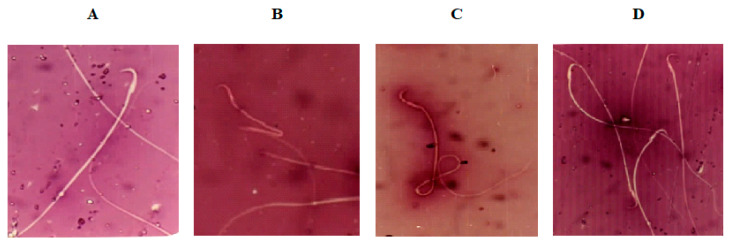
Photomicrographs (400×) of (**A**) normal sperm morphology of normal control (NC) rats; (**B**,**C**) sperm with bent midpiece and headless sperm, and sperm with coiled tail of diabetic control (DC) rats, respectively; and (**D**) normal sperm morphology of diabetic rats exposed to *Momordica charantia* at 500 mg/mg b.w.(D+MC-500 group).

**Figure 3 molecules-25-05255-f003:**
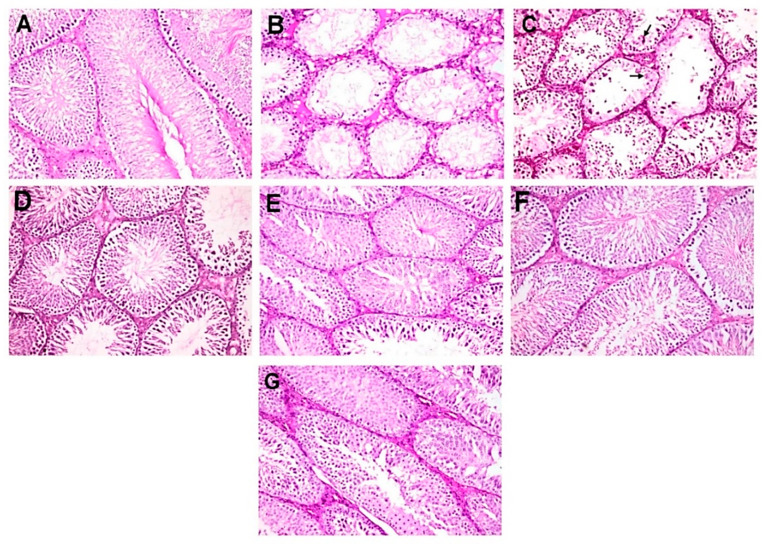
Photomicrographs of histopathological examination of testis (hematoxylin and eosin (H&E), 200×). (**A**) NC group showing normal histological picture of mature functioning seminiferous tubules. (**B**,**C**) DC group showing severe testicular degeneration with marked reduction or complete absence of spermatogenic cells with vacuolation of Sertoli cells, intertubular edema, and presence of spermatid giant cell (arrow). (**D**,**E**) D+MC-250 group showing restoration of normal architecture of seminiferous tubules with incomplete spermatogenic series. (**F**,**G**) D+MC-500 group showing restoration of normal architecture of seminiferous tubules with slight reduction in spermatogenic cells.

**Figure 4 molecules-25-05255-f004:**
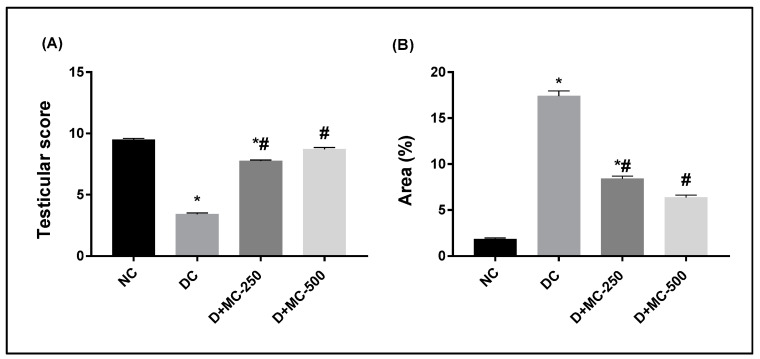
Bar charts of testicular lesion score and immunohistochemical analysis of anti-caspase-3 in experimental groups: (**A**) testicular lesion score and (**B**) caspase-3 immunopositive reaction expressed as area %. * Significantly different from NC group; # significantly different from DC group.

**Figure 5 molecules-25-05255-f005:**
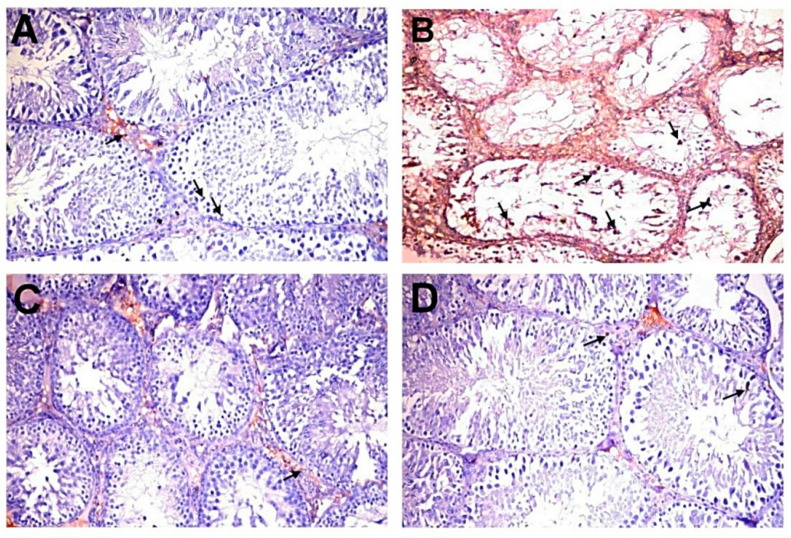
Immunohistochemical analysis of caspase-3 in experimental groups (200×). (**A**) NC group showing very weak immunopositive reaction (arrows). (**B**) DC group showing strong immunopositive reaction in germ, Sertoli (arrows), and Leydig cells. (**C**,**D**) D+MC-250 and D+MC-500 groups showing very weak immunoreactivity in spermatogenic (arrows) and Sertoli cells.

**Figure 6 molecules-25-05255-f006:**
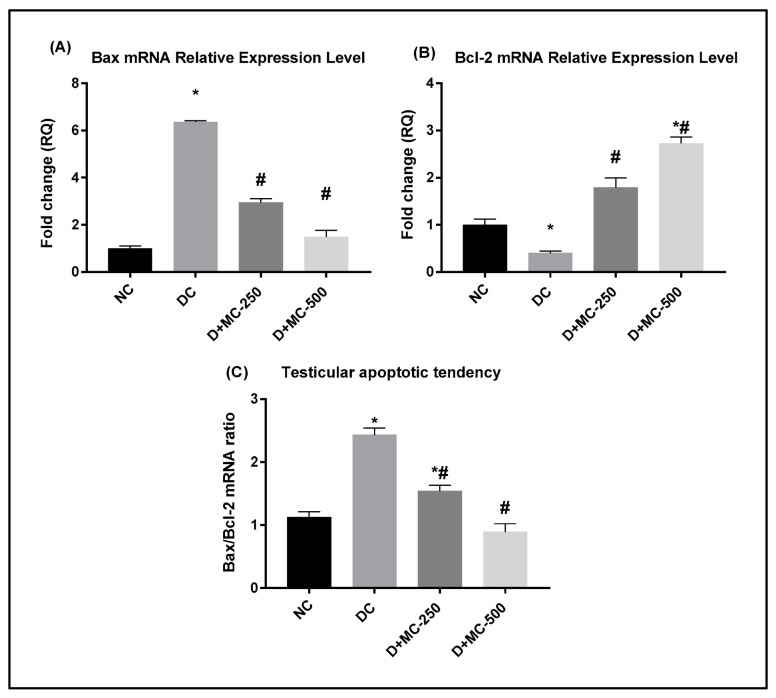
Effect of *Momordica charantia* (MC) extract against mitochondria-associated apoptotic cell death in testes of STZ-induced diabetic rats was examined by real-time RT-PCR: (**A**) relative Bax mRNA expression, (**B**) relative Bcl-2 mRNA expression, and (**C**) Bax mRNA to Bcl-2 mRNA ratio (Bax/Bcl-2). Data are presented as mean ± SEM. * Significantly different from NC group; # significantly different from DC group.

**Table 1 molecules-25-05255-t001:** Effect of *Momordica charantia* (MC) on blood levels of glucose, insulin, and glycosylated hemoglobin (HbA1c) of streptozotocin (STZ)-induced diabetic male rats. FBG, fasting blood glucose; NC, normal control; DC, diabetic control; D+MC-250, MC 250 mg/kg b.w. treatment; D+MC-500, MC 500 mg/kg b.w. treatment.

Group	FBG (mg/dL)		Insulin (mIU/L)		HbA1c (%)	
	0	12 w	0	12 w	0	12 w
NC	98.3 ± 2.46	99.5 ± 3.24	8.4 ± 0.30	8.2 ± 0.35	6.3 ± 0.21	6.2 ± 0.32
DC	375.8 ± 8.25 *	384.2 ± 7.43 *	4.7 ± 0.21 *	4.1 ± 0.27 *	6.5 ± 0.22	11.4 ± 0.46 *
D+MC-250	349.3 ± 7.27 *	135.5 ± 4.55 *#	4.7 ± 0.24 *	6.6 ± 0.31 *#	6.5 ± 0.20	7.4 ± 0.39 *#
D+MC-500	346 ± 13.89 *	113.2 ± 6.65 #	4.6 ± 0.25 *	7.2 ± 0.32 #	6.3 ± 0.24	6.7 ± 0.35 #

Values are expressed as mean ± SEM of six animals in each group. * Significantly different from values of normal control rats at *p* ≤ 0.05. # Significantly different from values of diabetic control rats at *p* ≤ 0.05.

**Table 2 molecules-25-05255-t002:** Effect of MC on antioxidant profile: superoxide dismutase (SOD), glutathione peroxide (GPx), catalase (CAT), glutathione (GSH), and malondialdehyde (MDA) in testicular homogenate of STZ-induced diabetic male rats.

Group	SOD (U/mg Protein)	GPx (U/mg Protein)	CAT (U/mg Protein)	GSH (μmol/g Tissue)	MDA (nmol/g Tissue)
NC	48.3 ± 1.56	6.9 ± 0.42	11.0 ± 0.57	10.9 ± 0.57	27.0 ± 0.70
DC	25.9 ± 0.53 *	1.4 ± 0.13 *	4.1 ± 0.29 *	4.0 ± 0.22 *	52.5 ± 1.37 *
D+MC-250	40.1 ± 1.08 *#	5.2 ± 0.30 *#	7.7 ± 0.35 *#	8.7 ± 0.44 *#	32.1 ± 1.02 *#
D+MC-500	45.8 ± 1.66 #	5.9 ± 0.25 #	9.7 ± 0.33 #	9.9 ± 0.29 #	25.9 ± 1.51 #

Values are expressed as mean ± SEM of six animals in each group. * Significantly different from values of normal control rats at *p* ≤ 0.05. # Significantly different from values of diabetic control rats at *p* ≤ 0.05.

**Table 3 molecules-25-05255-t003:** Effect of MC on blood levels of testosterone, follicle-stimulating hormone (FSH), and luteinizing hormone (LH) of STZ-induced diabetic male rats.

Group	TST (ng/mL)		FSH (mIU/mL)		LH (mIU/mL)	
	0	12 w	0	12 w	0	12 w
NC	2.62 ± 0.13	2.62 ± 0.17	4.57 ± 0.25	4.57 ± 0.29	0.75 ± 0.05	0.75 ± 0.04
DC	2.35 ± 0.11	0.95 ± 0.05 *	4.37 ± 0.21	1.73 ± 0.13 *	0.72 ± 0.03	0.18 ± 0.01 *
D+MC-250	2.45 ± 0.10	1.32 ± 0.11 *#	4.42 ± 0.25	2.77 ± 0.21 *#	0.73 ± 0.03	0.37 ± 0.03 *#
D+MC-500	2.52 ± 0.16	1.53 ± 0.12 *#	4.53 ± 0.27	2.95 ± 0.24 *#	0.74 ± 0.05	0.40 ± 0.03 *#

Values are expressed as mean ± SEM of six animals in each group. * Significantly different from values of normal control rats at *p* ≤ 0.05. # Significantly different from values of diabetic control rats at *p* ≤ 0.05.

**Table 4 molecules-25-05255-t004:** Effect of MC on sperm cell characteristics of STZ-induced diabetic male rats.

Group	Sperm Cell Characteristics			
	Count (×10^6^/mL)	Motility (%)	Viability (%)	Abnormalities (%)
NC	68.9 ± 2.64	88.5 ± 4.58	91.8 ± 4.72	5.9 ± 0.28
DC	31.4 ± 1.62 *	45.2 ± 2.61 *	40.3 ± 2.19 *	16.1 ± 0.71 *
D+MC-250	58.6 ± 2.47 *#	65.0 ± 3.73 *#	60.4 ± 3.10 *#	9.6 ± 0.62 *#
D+MC-500	63.5 ± 2.31 *#	71.4 ± 3.62 *#	67.6 ± 3.52 *#	8.5 ± 0.40 *#

Values are expressed as mean ± SEM of six animals in each group. * Significantly different from values of normal control rats at *p* ≤ 0.05. # Significantly different from values of diabetic control rats at *p* ≤ 0.05.

**Table 5 molecules-25-05255-t005:** Effect of MC on body weight of STZ-induced diabetic male rats.

Parameter	NC	DC	Diabetic	
			D+MC-250	D+MC-500
Initial weight (g)	194.5 ± 6.47	198.6 ± 7.25	195.8 ± 6.10	200.6 ± 7.58
Final weight (g)	263.5 ± 7.17	174.4 ± 5.82 *	239.7 ± 6.78 *#	259.4 ± 7.43 #
Weight gained/lost (g%)	35.48 ± 2.32	–12.19 ± 0.55 *	22.42 ± 1.18 *#	29.31 ± 1.97 #

Values are expressed as mean ± SEM of six animals in each group. * Significantly different from values of normal control rats at *p* ≤ 0.05. # Significantly different from the values of the diabetic control rats at *p* ≤ 0.05.

**Table 6 molecules-25-05255-t006:** Effect of MC on relative weight of sexual organs of STZ-induced diabetic male rats.

Group	Relative Weight (g) of Sexual Organs			
	Testes	Cauda Epididymis	Seminal Vesicles	Ventral Prostate
NC	1.32 ± 0.05	0.37 ± 0.03	0.28 ± 0.02	0.18 ± 0.01
DC	0.91 ± 0.03 *	0.22 ± 0.01 *	0.12 ± 0.01 *	0.07 ± 0.01 *
D+MC-250	1.14 ± 0.04 *#	0.29 ± 0.01 *#	0.20 ± 0.01 *#	0.13 ± 0.01 *#
D+MC-500	1.22 ± 0.04 #	0.34 ± 0.02 #	0.23 ± 0.02 #	0.15 ± 0.01 #

Values are expressed as mean ± SEM of six animals in each group. * Significantly different from values of normal control rats at *p* ≤ 0.05. # Significantly different from values of diabetic control rats at *p* ≤ 0.05.

**Table 7 molecules-25-05255-t007:** Effect of MC on mating success and male fertility % of STZ-induced diabetic male rats. CL, corpora lutea.

Groups	Number of Sperm-Positive Females	Mating Success (%)	Pregnancy Rate (%)	Number of Fetuses/Rat	Number of CL/Rat	Male Fertility (%)
NC	11	91.6	91.6	12.2 ± 0.56	13.2 ± 0.50	92.42 ± 1.71
DC	6	50.0	33.3	5.7 ± 0.27 *	8.8 ± 0.42 *	64.77 ± 1.92 *
D+MC-250	9	75.0	66.6	8.7 ± 0.43 *#	10.9 ± 0.46 *#	79.81 ± 1.85 *#
D+MC-500	10	83.3	83.3	9.8 ± 0.59 *#	11.6 ± 0.45 *#	84.48 ± 1.94 *#

Values are expressed as mean ± SEM of six animals in each group. * Significantly different from values of normal control rats at *p* ≤ 0.05. # Significantly different from values of diabetic control rats at *p* ≤ 0.05.

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
