# Peer review of "Momordica charantia Extract Protects against Diabetes-Related Spermatogenic Dysfunction in Male Rats: Molecular and Biochemical Study"

_molecules, 2020, doi:10.3390/molecules25225255_

Round 1

Reviewer 1 Report

Authors have clarified all the concerns raised. 

Minor comment: 

Please indicate the primer sequences of house keeping gene used for normalizing the data. 

Author Response

Please indicate the primer sequences of house keeping gene used for normalizing the data.

Done

Reviewer 2 Report

Summary of review:

The publication entitled Momordica charantia extract protects against diabetes-related spermatogenic dysfunction in male rats : molecular and biochemical study. By Maged aimed at investigating in an experimental rat model of diabetes the treatment effect of MC extract on the sperm cell characteristics, biochemical indicators of oxydative stress in homogenates, spermatogenic dysfunction, physiological biomarkers of diabetes (blood glucose, HbA1c, mating-success and male fertility test serum insulin, testostrone, FSH and LH hormes, with an testicular histology anlysis the protein expression levels of mitochondrial caspase-3 and by rtPCR apoptosis-related genes with relative Bcl-2 and Bax mRNA expressions. The experimental design used an animal model of STZ induced type 1 diabetes and 4 groups of analysis : a control and a diabetic groups and 2 diabetic groups with 250 and 500 mg/kg b.w. MC at the time of diabetes induction.

The main strengths are the following:

  • The effects of MC-treatment were assessed after an appropriate time i.e. 3 months of chronic exposure in an appropriate rat model well used for preclinical studies.
  • The results are correctly analysed and clearly presented.
  • A new aspect for testicular dysfunction during diabetes and extract plant treatment.
  • Two doses of MC plant extract have been used.

The main weaknesses and/or lacks are the following:

  • The effects of MC-treatment were investigated within 4 groups The study design would be more appropriated including a control, control + MC.
  • The STZ acts to induce as an oxydant, so an antioxydant is active, what is the main advantage for using MC ?
  • - replace mg /kg mg/kg b.w. (b.w. means body weight)
  • -in the discussion, a limitation section should be presented. As example the results presented here have been obtained in a type-1 model of diabetes so to extend to human and type-2, limitations should be presented. A preventive aspect of MC is pertinent but not a curative effect of MC since the introduction of MC should be after 2 months of diabetes. A similar preventive action to that presently found is of interest for curative action and so therapeutic interest.
  • A dose response relationship with the analysis of a third dose at a low dosage.

This paper is interesting for the impressive results, interesting effects and the mode of action at the testicular levels. Its pertinence for human use may be improved before publication.

Author Response

Summary of review:

The publication entitled Momordica charantia extract protects against diabetes-related spermatogenic dysfunction in male rats : molecular and biochemical study. By Maged aimed at investigating in an experimental rat model of diabetes the treatment effect of MC extract on the sperm cell characteristics, biochemical indicators of oxydative stress in homogenates, spermatogenic dysfunction, physiological biomarkers of diabetes (blood glucose, HbA1c, mating-success and male fertility test serum insulin, testostrone, FSH and LH hormes, with an testicular histology anlysis the protein expression levels of mitochondrial caspase-3 and by rtPCR apoptosis-related genes with relative Bcl-2 and Bax mRNA expressions. The experimental design used an animal model of STZ induced type 1 diabetes and 4 groups of analysis : a control and a diabetic groups and 2 diabetic groups with 250 and 500 mg/kg b.w. MC at the time of diabetes induction.

The main strengths are the following:

  • The effects of MC-treatment were assessed after an appropriate time i.e. 3 months of chronic exposure in an appropriate rat model well used for preclinical studies.
  • The results are correctly analysed and clearly presented.
  • A new aspect for testicular dysfunction during diabetes and extract plant treatment.
  • Two doses of MC plant extract have been used.

 The main weaknesses and/or lacks are the following:

  • The effects of MC-treatment were investigated within 4 groups The study design would be more appropriated including a control, control + MC.

Already exist, we used the first group as a normal control group, the second group as a diabetic control group, and the third and fourth groups as control + MC (250 and 500 mg/kg b.w., respectively).

  • The STZ acts to induce as an oxydant, so an antioxydant is active, what is the main advantage for using MC ?

Oxidative stress is not the only factor that controls STZ-induced diabetes. Hyperglycemia, is the responsible of the development of diabetes complications as well. Prolonged exposure to high glucose levels participates in the development of diabetic complications. Moreover, the development and progression of damage is proportional to hyperglycemia, which makes the lowering of glucose levels the most important goal for preventing complications of diabetes. Accordingly, the aim of this study was to explore the role of one of the antidiabetic plants against male sexual complications of diabetes.

  • - replace mg /kg mg/kg b.w. (b.w. means body weight)

Done

  • -in the discussion, a limitation section should be presented. As example the results presented here have been obtained in a type-1 model of diabetes so to extend to human and type-2, limitations should be presented. A preventive aspect of MC is pertinent but not a curative effect of MC since the introduction of MC should be after 2 months of diabetes. A similar preventive action to that presently found is of interest for curative action and so therapeutic interest.

This study is not directed at treating any type of diabetes, as we used a plant with known anti-diabetic effect so we did not suggest that this study would extend to human or type-2 diabetes. Further, we explained that the study was aimed at assessing the potential effect of M. charantia extract on protection against diabetes-related sexual complications in male rats.

The title of the study (Momordica charantia extract protects against diabetes-related spermatogenic dysfunction in male rats: Molecular and biochemical study) and the conclusion confirmed our aim. We mentioned in conclusion that the study indicates that MC is a possible new protective agent for diabetes-related male sexual complications.

According to the study design, we suggested that the extract be used for prevention, not treatment, and this is what we mentioned in the title of the manuscript and in the conclusion.

In the last sentence of conclusion, we modified the word "therapeutic" to "protective" to be in line with the reviewer’s comment.

  • A dose response relationship with the analysis of a third dose at a low dosage.

The selection of both doses used in this study was based on our previous study “Molecular and biochemical monitoring of the possible herb-drug interaction between Momordica charantia extract and glibenclamide in diabetic rats. Saudi Pharm. J. 2019, 27(6), 803-816”.

This paper is interesting for the impressive results, interesting effects and the mode of action at the testicular levels. Its pertinence for human use may be improved before publication.

Round 2

Reviewer 2 Report

The manuscript entitled “Momordica charantia extract protects against diabetes-related spermatogenic dysfunction in male rats : molecular and biochemical study ». By Maged et al. (sorry for Soliman/M) has been improved.

I have the following comments following the reading of answers from the cover letter.

The experimental design used an animal model of STZ induced type 1 diabetes and 4 groups of analysis : a control and a diabetic groups and 2 diabetic groups with 250 and 500 mg/kg b.w. MC at the time of diabetes induction.

My question was to discuss the presence of a control group (without diabetes) with the application of Momordica charantia extract treatment.

The answer given by the authors is not appropriated to me. You should give the reasons to do not have this control group. Have you historical reasons ?

  • To my question: A dose response relationship with the analysis of a third dose at a low dosage.

From a pharmacological point of view, two doses of 250 mg and 500 mg have been used but are of little importance for the pharmacological definition of a mode of action. This 2 doses are too closed for an animal study (for human it is ok). The paper would be acceptable with one of the two doses used here. My suggestion for the future is to use 3 doses instead 2 is to document the dose response relationship in an animal model as a preclinical study by a logarithmic dose relation (for a pharmacological aspect). For example, we have this relationship with a dose of 50 mg as referred to the maximal dose of 500 mg. This should be discussed.

The general design for such studies is to have a Control group (C), a group with M. charancia (C+Mc) a diabetic control group (D) and a diabetic treated group with one doe of Mc (D+Mc). In the table only MC-250 or 500 is presented. This should be replaced by D+MC-250 and D+MC-500. The implicite question is the next step with the clinical application : is the dose of Mc applicable for human treatment ? The answer would be yes as it has been cited by the reference 10 (clinical trials). This should be discussed in the present manuscript.

To my request:Its pertinence for human use may be improved before publication.

So a limitation point should be included in the discussion explaining the lack of a control group with Mc and the minimum dose useful for the clinical purpose.

Author Response

My question was to discuss the presence of a control group (without diabetes) with the application of Momordica charantia extract treatment.

The answer given by the authors is not appropriated to me. You should give the reasons to do not have this control group. Have you historical reasons?

We did not use a non-diabetic control group treated with extract, as we were satisfied with comparing the results of the study with the normal control and diabetic control groups. But, based on the recommendations of the reviewers, we mentioned this group as one of the study limitations.

  • To my question:A dose response relationship with the analysis of a third dose at a low dosage.

From a pharmacological point of view, two doses of 250 mg and 500 mg have been used but are of little importance for the pharmacological definition of a mode of action. This 2 doses are too closed for an animal study (for human it is ok). The paper would be acceptable with one of the two doses used here. My suggestion for the future is to use 3 doses instead 2 is to document the dose response relationship in an animal model as a preclinical study by a logarithmic dose relation (for a pharmacological aspect). For example, we have this relationship with a dose of 50 mg as referred to the maximal dose of 500 mg. This should be discussed.

We thank the reviewers for this note, which we will follow in our future studies.

The general design for such studies is to have a Control group (C), a group with M. charancia (C+Mc) a diabetic control group (D) and a diabetic treated group with one doe of Mc (D+Mc). In the table only MC-250 or 500 is presented. This should be replaced by D+MC-250 and D+MC-500.

Done and the groups names were changed.

The implicite question is the next step with the clinical application: is the dose of MC applicable for human treatment? The answer would be yes as it has been cited by the reference 10 (clinical trials). This should be discussed in the present manuscript.

Done as recommended.

To my request: Its pertinence for human use may be improved before publication.

So a limitation point should be included in the discussion explaining the lack of a control group with Mc and the minimum dose useful for the clinical purpose.

Done as recommended.

This manuscript is a resubmission of an earlier submission. The following is a list of the peer review reports and author responses from that submission.

Round 1

Reviewer 1 Report

Soliman et al, studied the effect if Momordica charantia extract against spermatogenic dysfunction. Various reports suggest that M. charantia is known to have anti-diabetic effect as used in traditional medicine. Authors, in this study trying to describe the effect of M. charantia extract in the diabetes induced reproductive complications in-vivo. For this, authors have designed beautiful study with thorough methodological flowchart starting from extraction, the role of extract in decreasing blood sugar, effect on oxidative stress in testicular tissue, effect on hormonal ablation (TST and Gonadotropins) and leading to changes in sperm characteristic and finally addressing the mechanism via apoptosis by assessing Bcl family genes and Caspase-3. However, lack of evidences suggest that authors failed to execute their study. Below mentioned are some serious flaws need to be rectified and hence it is not suitable for publication in Molecules at this stage.  The concerns mentioned below may help authors to revise and completely reorganize the article for consideration in future.  

Major concerns:

  • As the starting material for this study, authors have not explained well about the preparation of extracts though they cite their previous study (Soliman et al 2019), in which they have performed leaf extraction. In the current study they have used fresh fruits of M charantia, and details regarding the amount of starting material they used and the quantity of extract they got are completely missing. Authors needs to provide simple biochemical evidences about the phytochemical constituents present in their crude extract to convince the effect of extracts are due to these active phytochemical constituents.
  • Though authors perform LC-MS study and the results are poorly described (big data set in just two lines!!!!! Page 5 line 221-223). I agree that they can cite their previous paper in method section but still in results section they cannot just cite previous paper. If the results for this article published previously in other journal, then there is no point of newly submitting the results to new journal. I believe the results of M. charantia extracts needs to be elaborated to describe clearly in ‘THIS ARTICLE’. The conditions, apparatus etc should be given in method section and results described properly. What are the percentage of the molecular peaks??
  • Figure 1 is unclear, legend says chemical structure of Momordicine I and Momordicin II but I could see the structure of one compound and I am not sure whether it is Momordicine I or Momordicin II.  
  • Two concentrations of the extracts have been selected for the study. There is no rationale for selecting the dose. No toxicity studies have been performed. LD-50 have not been performed to select the doses. There should be a rationale to select doses, being a reader, one should know why the indicated doses have been used. What will happen if MC-750 used, animals will die??
  • Figure two has three panels, A. from control rats and B&C from diabetic rats??? It is unclear. If showing control and diabetic, as a reader I would like to see the Images from MC treated rats as well.
  • H&E (Fig 3) and Immunohistochemistry of Caspase-3 (Fig 5) images looks good, but scale bar needs to be indicated.
  • I have serious issue with the mechanistic part of the study. Authors have used qPCR approach to explain the efficacy of extract in inhibiting apoptosis. They used Bcl-2 and Bax primers. They have mentioned these list of primers Bax 5′- GAGGATTGTGGCCTTCTTTG-3′ and 5′-CGTTATCCTGGATCCAGGTG-3′ and Bcl-2 5′- ACCAAGAAGCTGAGCGAGTG-3′, reverse: 5′-CCAGTTGAAGTTGCCGTCTG-3. I did a quick nBLAST just to verify the accuracy of primers listed and primers did not show homology with the mentioned primes. Bax enlisted primers showed homology with Bcl-2 primers abd Bcl-2 primers showed homology with Bax primers. If this is the case, then the results mentioned in figure 6 are contradictory to what they are trying to explain. By the way, the authors have cited a paper for qPCR methodology (Ashraf et al, 2018), in which primers and their naming are matching. If it is not a typo, the authors need to be justified their results and have to change their whole hypothesis.
  • I found lot of typos which shows authors are in too much hurry to publish.  

Reviewer 2 Report

The manuscript in title of  Momordica charantia extract protects against 2 diabetes-related spermatogenic dysfunction in male 3 rats: Molecular and biochemical study aims to explore the biological effect of Momordica charantia by  estimating the blood levels of insulin, glucose, 23 glycosylated hemoglobin (HbA1c), testosterone (TST), follicle stimulating hormone (FSH) and 24 luteinizing hormone (LH). The authors reported that MC resulted in a significant reduction of blood glucose and HbA1c and 30 marked elevation of serum levels of insulin, TST and gonadotropins in diabetic rats. This is an interesting study and the manuscript is well organized. However, there are several issues needed be fixed before it could be accepted for publication.

  1. In the Figure 3. Photomicrograph of histopathological examination of the testis. Please use comparable images;
  2. The Fig 4 the resolution of Fig 4 is not acceptable, please provide image > 300dpi.
  3. In the Figure 5. Immunohistochemical analysis of caspase-3 in the different experimental groups, please double check positive signals of caspase-3.
  4. The Fig 6 the resolution of Fig 4 is not acceptable, please provide image > 300dpi.
  5. More information about the male fertility test.
  6. In the section of “Funding: Please add: This research was funded by Deanship of Scientific Research (DSR), Prince Sattam bin 537 Abdulaziz University, Al-Kharj, Saudi Arabia.”. There is typo “please add”.